🔓 | **Open Peer Review** | Clinical Microbiology | Research Article

# Elevating fungal care: bridging Brazil's healthcare practices to global standards

Jon Salmanton-García,[1,2,3] Diego R. Falci,[4,5] Oliver A. Cornely,[1,2,3,6] Alessandro C. Pasqualotto,[7,8] IFI Diagnostic and Treatment Capacity Teams Brazil and Europe

**ABSTRACT** Brazil faces unique challenges in managing invasive fungal infections (IFIs) due to diverse ecosystems, a rural workforce, and prevalent health conditions. In Europe, IFIs are primarily associated with transplantation, intensive care, and chronic diseases. Inspired by initiatives in the Caribbean and Latin America in 2019, efforts to map global diagnostic and treatment resources expanded to Africa, Europe, and Asia/Pacific. This study conducts a comparative analysis, mainly drawing data from Brazil and Europe, to investigate IFI epidemiology and management. Data were collected through online surveys distributed to Brazilian and European institutions, with collaborations from scientific organizations. Surveys covered institutional profiles, IFI diagnoses, accessibility to diagnostic techniques, and antifungal drugs. A comparative survey involving 96 Brazilian and 388 European institutions revealed variations in the perception and practices related to fungal pathogens. Differences in ranking and prevalence were observed, along with variations in diagnostic procedures, fluorescence dye usage, culture practices, antifungal medication availability, and technological approaches. Europe exhibited higher utilization rates for molecular diagnostic approaches, including PCR tests, and therapeutic drug monitoring (TDM) was more widespread in Europe compared with Brazil, indicating substantial differences in understanding and managing fungal infections. Customized IFI management is crucial, considering regional differences and addressing technological gaps like underutilized PCR. The study advocates for increased international collaboration, targeted training, and enhanced resources to foster a unified global approach in preventing, diagnosing, and treating IFI.

**IMPORTANCE** This work is significant as it highlights the unique challenges Brazil faces in managing invasive fungal infections (IFIs) due to its diverse ecosystems and public health landscape. By comparing Brazil's situation with Europe—where IFIs are mainly linked to transplantation and intensive care—this study identifies key disparities in diagnostic and treatment practices. The findings reveal substantial differences in the availability and use of molecular diagnostics, antifungal drugs, and therapeutic drug monitoring, with Europe demonstrating more advanced practices. By mapping these variations, the study underscores the importance of tailored approaches to IFI management that consider regional differences and technological gaps. Ultimately, it calls for enhanced international collaboration, targeted training, and resource allocation to improve IFI outcomes globally, particularly in regions with limited access to advanced diagnostic tools and treatments.

**KEYWORDS** invasive fungal infections, Brazil, Europe, diagnosis, antifungals, PCR, MALDI-TOF-MS

**Peer Reviewer** Ariful Basher, Infectious diseases and Tropical Medicine Department, Mymensingh Medical College and Hospital, Mymensingh, Bangladesh

Address correspondence to Jon Salmanton-García, jon.salmanton-garcia@uk-koeln.de.

Jon Salmanton-García and Diego R. Falci contributed equally to this article. The author order was based on amount of contribution and ulterior randomisation of names.

Oliver A. Cornely and Alessandro C. Pasqualotto are joint senior authors.

The authors declare no conflict of interest.

Brazil, the largest country in South America, boasts a population of nearly 220 million people (1) spread across a territory of over 8.5 million km$^2$ (2). The susceptibility to

invasive fungal infections (IFIs) stems from several factors, including exposure to diverse ecosystems that harbor pathogenic fungi (3), including *Coccidioides* spp., *Histoplasma* spp., *Paracoccidioides* spp., and *Sporothrix* spp. (3–6), a significant rural workforce, and elevated rates of HIV/AIDS (4, 7–9), tuberculosis (10), malignancies (11), and solid organ (12) and hematopoietic stem cell transplantations. (4, 13)

On the other hand, in Europe, a continent with a population of over 740 million people spread across an area of over 10 million km$^2$ (14), IFIs are primarily associated with malignancy, solid organ and haematopoietic stem cell transplantations, intensive care, and chronic diseases (*e.g.*, heart-, kidney-, or liver-related) (7, 15, 16). Nevertheless, the Eastern region of the continent experiences a higher prevalence of HIV/AIDS and tuberculosis, exposing individuals to specific fungal pathogens (17–19). Unlike Brazil, Europe rarely encounters endemic mycoses, with most cases being imported (20–23). However, certain regions in Southern Europe face an increased risk of becoming endemic zones for specific pathogenic fungi (24) and other pathogens due to climate change. (25)

The importance of mapping access to diagnostic and treatment resources for IFI gained recognition following insights from local experiences in the Caribbean and Latin America in 2019 (26). This spurred similar initiatives in other continents, including Africa (27), Europe (28, 29), and Asia/Pacific (30). Additionally, spin-off efforts have been initiated or are underway in specific countries or regional clusters (31–34). This comparative analysis relies heavily on data from Brazil, which contributed to over 90% of the centers analyzed in the Caribbean and Latin America (26). Furthermore, Europe (29), acknowledged as a continent with most advanced capabilities for diagnosing and treating IFI, plays a crucial role as a data source.

The scientific analysis comparing IFI management practices in Brazil and Europe, as the region where highest standards have been shown, offers Brazil valuable insights for healthcare enhancement. By benchmarking against Europe, Brazil can identify best practices, address technological gaps in diagnostic capabilities, and improve access to antifungal medications. Understanding climate change implications enables Brazil to prepare for emerging health challenges. The review aims to inform policy recommendations, capacity building initiatives, and opportunities for further research. Brazil's active participation in global health initiatives enhances its standing in the international healthcare community, while the study's focus on disparities and regional variations provides a roadmap for targeted improvements in diagnostic infrastructure, public health awareness, and overall healthcare outcomes.

## MATERIALS AND METHODS

Data were collected utilizing a single online case report form. Scientists from all Brazilian federal states and European countries were invited to participate, extending invitations not only to the authors' immediate collaborators but also to members of national, regional, and international scientific organizations. Additionally, calls for participation were disseminated across various online platforms. Data for Brazil were collected between February to September 2018 after disseminating invitations through the Leading International Fungal Education (LIFE) initiative, Brazilian Society of Infectious Diseases (SBI), Pan American Infectiology Association (API), Brazilian Society of Clinical Analysis (SBAC), and Brazilian Society of Microbiology (SBM). For Europe, the survey was distributed through the International Society of Human and Animal Mycology (ISHAM) and the European Confederation of Medical Mycology (ECMM), and data collected between 1 November 2021 and 31 January 2022. No informed consent was applicable to this research. No animal experiments are included in this research. Individual data for Brazil and Europe were published separately in 2019 and 2023, respectively (26, 29)

Before analysis, the two datasets were combined into a unified unit, ensuring meticulous oversight for both completeness and consistency. The surveys covered a range of topics, including i) institutional profiles, ii) the frequency of invasive fungal infection (IFI) diagnoses, iii) accessibility to microscopy, cultural methods, antibody and

antigen detection kits, and molecular assays, and iv) availability of antifungal drugs and therapeutic drug monitoring (TDM). Participants were required to indicate "available" or "not available" for each technique at their institutions. The overall perceived IFI incidence was assessed on a Likert scale from 1 (very low) to 5 (very high).

The presentation of data involved organizing frequencies, percentages, and proportions into contingency tables. Statistical proportion comparisons were performed using the χ test, considering a significance level of $P < 0.05$ as statistically significant. Analyses were exclusively conducted on valid responses in the respective summaries or comparisons, with centers lacking responses being excluded. The statistical analyses were carried out using SPSS v27.0 (SPSS, IBM Corp., Chicago, IL, United States).

## RESULTS

A comparative analysis of microbiology laboratory practices and mycological diagnostic performance in Brazil and Europe revealed notable disparities across various aspects. The Brazilian survey involved 96 institutions (0.4/million inhabitants), and the European survey had 388 institutions (0.5/million inhabitants).

In terms of operational structure, 66.7% of analyzed sites in Brazil had onsite microbiology laboratories, contrasting with Europe where a significantly higher majority (94.8%, $P < 0.001$) had such facilities. A detailed examination of mycological diagnostic practices revealed differences in the capacity to perform them in-house, with 44.8% in Brazil and 58.0% in Europe, as well as hybrid onsite–outsourced models (24.0% in Brazil vs 37.4% in Europe, $P < 0.001$) (Table 1).

Significant differences emerged in the perception of major fungal pathogens. *Aspergillus* spp. ranked higher in Europe (88.9%) than in Brazil (47.9%, $P < 0.001$). Similar trends were observed for *Candida* spp. (Europe 94.3% vs Brazil 90.6%, $P < 0.001$) and Mucorales (Europe 29.9% vs Brazil 8.3%, $P < 0.001$). Conversely, *Cryptococcus* spp. was perceived as more relevant in Brazil (62.5%) than in Europe (22.7%, $P < 0.001$), so as *Histoplasma* spp. (Brazil 44.8% vs Europe 4.1%, $P < 0.001$). *Fusarium* spp. showed similar relevance in both regions (Brazil 19.8% vs Europe 21.6%, $P = 0.690$) (Fig. 1).

In terms of diagnostic procedures, microscopy was highly utilized in both regions (Brazil 95.8% vs Europe 96.6%, $P = 0.697$). However, specific techniques like optical brighteners differed significantly (Brazil 5.2% vs Europe 46.4%, $P < 0.001$). Fluorescence dye usage was more prevalent in Europe (62.6%) than in Brazil (17.7%). Culture practices varied, with 94.8% of Brazilian sites performing cultures compared with 98.7% in Europe ($P < 0.001$). Access to certain culture methods also showed significant differences, such as potato dextrose agar, Sabouraud dextrose agar + chloramphenicol, Sabouraud dextrose agar + gentamicin, and selective agar ($P < 0.001$). Methods for species identification, in particular automated identification (Brazil 55.2% vs Europe 59.3%, $P = 0.723$), showed similarity between the two regions, except for MALDI-TOF-MS, which was more commonly used in Europe (74.0%) compared with Brazil (14.6%) ($P < 0.001$). Regarding technologies for antifungal susceptibility testing, disparities existed in access to broth microdilution with the EUCAST standard (Brazil 5.2% vs Europe 42.5%, $P < 0.001$) and gradient strip tests (Brazil 24.0% vs Europe 59.5%, $P < 0.001$). Overall access to at least one antibody detection kit (Brazil 46.9% vs Europe 82.2%) and specific access to antibody testing for *Aspergillus* spp. (Brazil 32.3% vs Europe 78.4%) differed significantly ($P < 0.001$), but not in the case of the test for *Histoplasma* spp. (Brazil 38.5% vs Europe 45.6%, $P = 0.211$). Access to all antigen testing methods was notably higher in Europe than in Brazil ($P < 0.001$). Regarding molecular diagnostic approaches, particularly polymerase chain reaction (PCR) tests, in Brazil, PCR tests for *Aspergillus*, *Candida*, and *Pneumocystis* were employed by 9.4%, 11.5%, and 12.5%, respectively. In contrast, Europe exhibited substantially higher utilization rates, with 66.0%, 54.1%, and 74.2%, respectively ($P < 0.001$) (Table 1).

Significant discrepancies in the availability of antifungal medications were also observed between Brazil and Europe. Amphotericin B, in any formulation, was notably more prevalent in Europe (86.9%) than in Brazil (71.9%) ($P < 0.001$). However,

**TABLE 1** Baseline characteristics and access to laboratory tools of participating institutions in Brazil and Europe[a]

| | Brazil | | Europe | | P value |
|---|---|---|---|---|---|
| | n | % | n | % | |
| Microbiology laboratory | | | | | <0.001 |
| Onsite | 64/96 | 66.7 | 368/388 | 94.8 | |
| Outsourced | 26/96 | 27.1 | 19/388 | 4.9 | |
| Mycological diagnosis performance | | | | | <0.001 |
| Onsite | 43/96 | 44.8 | 225/388 | 58.0 | |
| Onsite–outsourced | 23/96 | 24.0 | 145/388 | 37.4 | |
| Outsourced | 22/96 | 22.9 | 13/388 | 3.4 | |
| Microscopy | 92/96 | 95.8 | 375/388 | 96.6 | 0.697 |
| Methodologies | | | | | |
| Calcofluor white | 5/96 | 5.2 | 180/388 | 46.4 | <0.001 |
| Giemsa stain | 46/96 | 47.9 | 210/388 | 54.1 | 0.275 |
| China/India ink | 77/96 | 80.2 | 303/388 | 78.1 | 0.651 |
| Potassium hydroxide | 52/96 | 54.2 | 223/388 | 57.5 | 0.558 |
| Silver stain | 37/96 | 38.5 | 147/388 | 37.9 | 0.906 |
| Access to fluorescence | 17/96 | 17.7 | 243/388 | 62.6 | <0.001 |
| Direct examination if cryptococcosis suspected | 88/96 | 91.7 | 319/388 | 82.2 | 0.023 |
| Silver stain if *Pneumocystis pneumonia* suspected | 34/96 | 35.4 | 120/388 | 30.9 | 0.398 |
| Direct microscopy if mucormycosis suspected | NR | NR | 211/388 | 54.4 | —[b] |
| Culture | 91/96 | 94.8 | 383/388 | 98.7 | 0.016 |
| Fungal culture methods | | | | | |
| Agar Niger | 18/96 | 18.8 | 46/388 | 11.9 | 0.074 |
| Chromogen | NR | NR | 187/388 | 48.2 | — |
| Lactrimel agar | 3/96 | 3.1 | 31/388 | 8.0 | 0.095 |
| Potato dextrose agar | 19/96 | 19.8 | 148/388 | 38.1 | <0.001 |
| Sabouraud dextrose agar | 73/96 | 76.0 | 293/388 | 75.5 | 0.914 |
| Sabouraud dextrose agar + Chloramphenicol | 27/96 | 28.1 | 245/388 | 63.1 | <0.001 |
| Sabouraud dextrose agar + Gentamicin | 9/96 | 9.4 | 175/388 | 45.1 | <0.001 |
| Selective agar | 21/96 | 21.9 | 207/388 | 53.4 | <0.001 |
| Available tests for specific identification | 78/96 | 81.3 | 372/388 | 95.9 | <0.001 |
| Automated identification | 53/96 | 55.2 | 230/388 | 59.3 | 0.723 |
| Biochemical tests | 59/96 | 61.5 | 208/388 | 53.6 | 0.166 |
| DNA sequencing | NR | NR | 187/388 | 48.2 | — |
| MALDI-TOF-MS | 14/96 | 14.6 | 287/388 | 74.0 | <0.001 |
| Mounting medium | NR | NR | 113/388 | 29.1 | — |
| Antifungal susceptibility technologies | 57/96 | 59.4 | 366/388 | 94.3 | <0.001 |
| CLSI | 20/96 | 20.8 | 106/388 | 27.3 | 0.195 |
| EUCAST | 5/96 | 5.2 | 165/388 | 42.5 | <0.001 |
| Gradient strip test | 23/96 | 24.0 | 231/388 | 59.5 | <0.001 |
| Semiautomated antifungal susceptibility testing system | 35/96 | 36.5 | 143/388 | 36.9 | 0.942 |
| Antibody detection | 45/96 | 46.9 | 319/388 | 82.2 | <0.001 |
| *Aspergillus* spp. | 31/96 | 32.3 | 304/388 | 78.4 | <0.001 |
| *Candida* spp. | NR | NR | 239/388 | 61.6 | — |
| *Histoplasma* spp. | 37/96 | 38.5 | 177/388 | 45.6 | 0.211 |
| Antigen detection | 60/96 | 62.5 | 363/388 | 93.6 | <0.001 |
| *Aspergillus* galactomannan, any | 30/96 | 31.3 | 351/388 | 90.5 | <0.001 |
| *Candida* mannan | NR | NR | 195/388 | 50.3 | — |
| *Cryptococcus* glucuronoxylomannan, any | 54/96 | 56.3 | 308/388 | 79.4 | <0.001 |
| *Histoplasma* antigen | 13/96 | 13.5 | 133/388 | 34.3 | <0.001 |
| Beta-d-glucan | 9/96 | 9.4 | 236/388 | 60.8 | <0.001 |

(*Continued on next page*)

**TABLE 1** Baseline characteristics and access to laboratory tools of participating institutions in Brazil and Europe[a] (Continued)

|  | Brazil | | Europe | | P value |
|---|---|---|---|---|---|
|  | n | % | n | % |  |
| Molecular tests | 18/96 | 18.8 | 329/388 | 84.8 | <0.001 |
| *Aspergillus* PCR | 9/96 | 9.4 | 256/388 | 66.0 | <0.001 |
| *Candida* PCR | 11/96 | 11.5 | 210/388 | 54.1 | <0.001 |
| *Pneumocystis* PCR | 12/96 | 12.5 | 288/388 | 74.2 | <0.001 |
| Mucorales PCR | NR | NR | 182/388 | 46.9 | – |

[a]CLSI, Clinical and Laboratory Standards Institute; DNA, deoxyribonucleic acid; ELISA, enzyme-linked immunosorbent assay; EUCAST, European Committee on Antimicrobial Susceptibility Testing; MALDI-TOF-MS, matrix-assisted laser desorption/ionization time-of-flight mass spectrometry; NR, not reported; PCR, polymerase chain reaction. Data from Brazil were previously included in a collective publication alongside data from other Caribbean and Latin American countries (26). The European data, in its present format, have been published separately elsewhere. (29).
[b]"–", not applicable.

no substantial difference was noted for amphotericin B deoxycholate ($P = 0.157$) or amphotericin B lipid complex ($P = 0.401$). Liposomal amphotericin B did show a significant contrast, with a more extensive availability in Europe (77.6%) compared with Brazil (34.4%) ($P < 0.001$). Disparities were also evident in the use of echinocandins, with a higher prevalence in Europe (89.2%) than in Brazil (55.2%) ($P < 0.001$). Accessibility to at least one triazole did not exhibit an overall significant difference ($P = 0.945$). However, while access to fluconazole was similar in both Brazil and Europe, mold-active triazoles were more frequently available in Europe ($P < 0.001$). Significant differences in the prevalence of therapeutic drug monitoring (TDM) were likewise observed between Brazil and Europe. Overall, access to TDM for at least one drug was markedly more widespread in Europe (64.4%) than in Brazil (9.4%) ($P < 0.001$). This notable difference persisted when examining individual antifungals (Fig. 2; Table S1).

## DISCUSSION

The manuscript conducts a comparative examination of IFI diagnostic and treatment practices in Brazil and Europe, regions marked by distinctive ecological, epidemiological, and healthcare characteristics. It sheds light on the varied factors influencing IFI susceptibility, including Brazil's exposure to specific fungal pathogens and a substantial rural workforce, juxtaposed against Europe's predominant associations with transplantation, intensive care, and chronic diseases (*e.g.*, heart-, kidney-, or liver-related). Noteworthy, discrepancies emerge in the accessibility of cutting-edge diagnostic techniques, encompassing cultivation methods, antigen and antibody test detection, and access to molecular assays—all of which are more prevalent in Europe than in Brazil. Moreover, substantial differences are evident in the availability of all antifungals, except fluconazole, deoxycholate amphotericin B, and amphotericin B lipid complex.

The study reveals substantial variations in microbiology laboratory practices between the two regions. Europe, with a higher proportion of onsite microbiology laboratories, showcases a more advanced infrastructure compared with Brazil (26, 29). Differences in mycological diagnostic practices, including the prevalence of onsite and outsourced models, shed light on the diverse approaches to fungal diagnosis in these regions.

Significant disparities emerge in the perception of major fungal pathogens. *Aspergillus* spp., *Candida* spp., and Mucorales are considered more relevant in Europe, whereas *Cryptococcus* spp. and *Histoplasma* spp. are perceived as more significant in Brazil. This highlights regional variations in fungal epidemiology, including the endemicity of *Histoplasma* spp. in Brazil (3–6), and underscores the need for tailored diagnostic and treatment strategies. In parallel, the ranking of fungal pathogens made by the participants in both studies appears to be similar to the relevance classification obtained in the World Health Organization's (WHO) fungal pathogen priority list. (35)

The study dissects diagnostic procedures, revealing both similarities and differences. While microscopy is commonly utilized in both regions, distinct techniques, such as

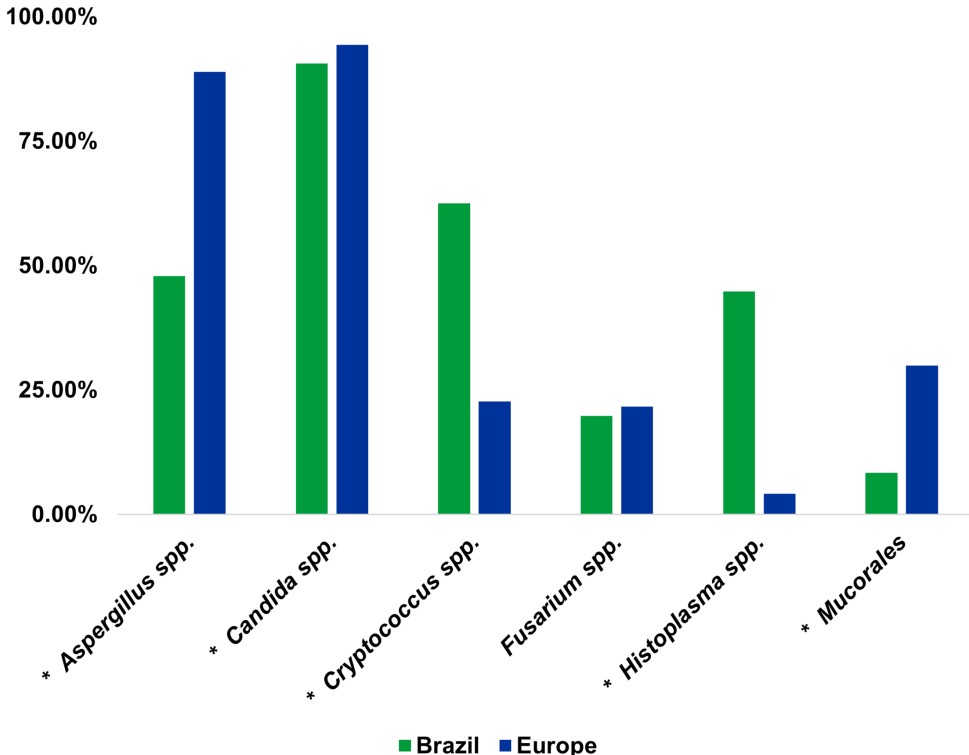

**FIG 1** Fungal pathogens described as having the most relevance in Brazil and Europe. *, $P < 0.05$ - χ test based; spp., species. Data from Brazil were previously included in a collective publication alongside data from other Caribbean and Latin American countries (26). The European data, in its present format, have been published separately elsewhere (29).

optical brighteners and fluorescence dye, exhibit notable regional differences. This is highly significant, as these tools contribute to heightened diagnostic sensitivity. The absence of these tools may lead to delays in diagnosis. Furthermore, variations in cultivation practices and access to culture methods further emphasize the diversity in diagnostic approaches. As outlined in previous experiences, such as the reports on the diagnostic and treatment capabilities for IFI in Africa and the Asia-Pacific region, the predominant factor accounting for this limitation in accessing specific diagnostic tools could be the costs (29, 30, 36, 37)

The technological gap between Brazil and Europe is evident, particularly in molecular diagnostic approaches. The significantly lower utilization of PCR tests in Brazil, compared with Europe, underscores the need for technology transfer and capacity building in the former, as this might be tightly related to the economical power. Advancements in the availability of specific PCR tests could have a significant impact, for instance addressing azole-resistant aspergillosis, which has already been recognized as a concern in Brazil, with a prevalence of at least 3% (38, 39). Variations in access to advanced methods, such as MALDI-TOF-MS for species identification, highlight areas for improvement in Brazil. Ultimately, utilizing these advanced techniques could potentially lead to a shortened diagnostic time, aligning with the earlier mention of optical brighteners and fluorescence. Additionally, unlike many European institutions that have access to all essential diagnostic tools outlined in the WHO list of essential *in vitro* diagnostics (40), the majority of Brazilian counterparts lack such comprehensive access, mirroring the circumstances observed in Africa and the Asia-Pacific region.

Differences in the obtainability of antifungal medications are apparent, with Europe generally exhibiting higher accessibility rates. Variations in the usage of specific antifungal drugs, including liposomal amphotericin B, echinocandins, and mold-active triazoles, point towards potential differences in treatment strategies and susceptibility patterns.

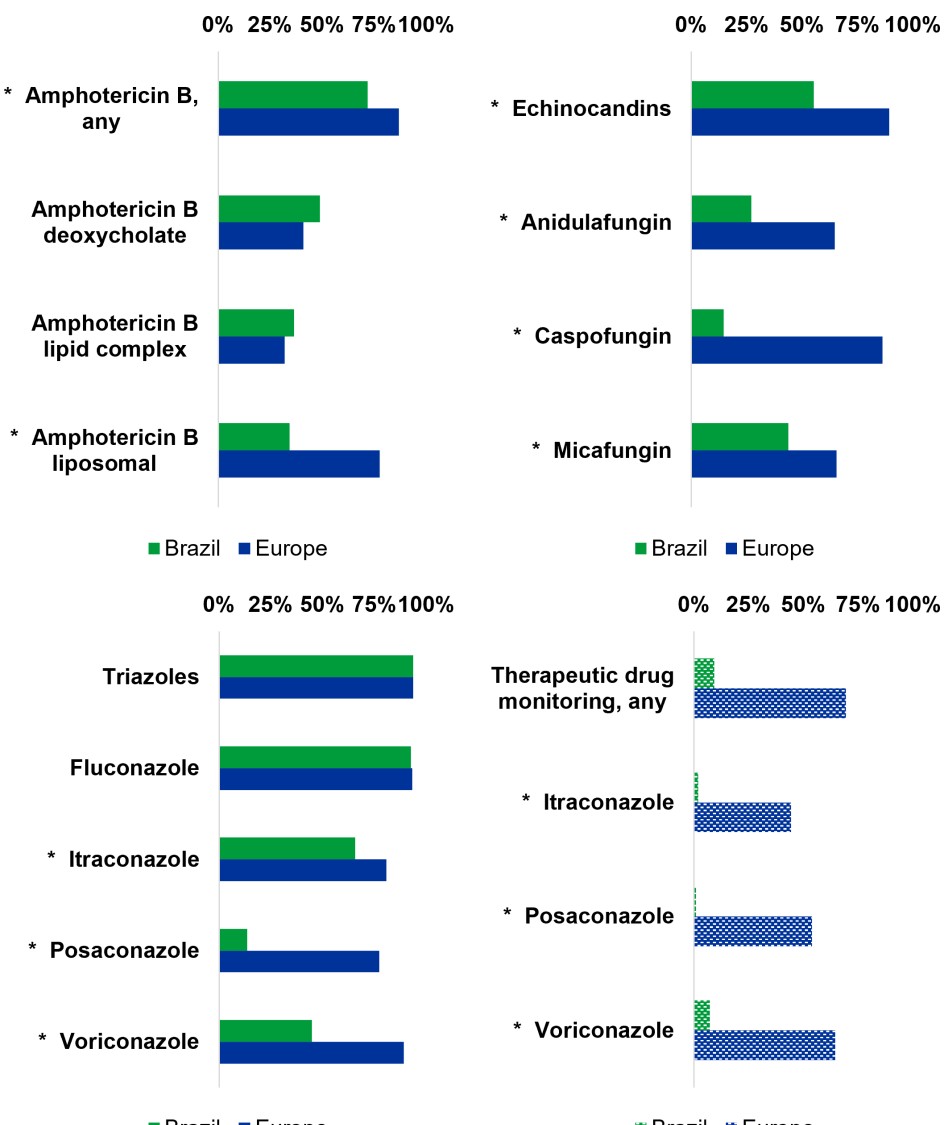

FIG 2   Access to antifungal treatments in participating institutions in Brazil and Europe. No information regarding the access to flucytosine, isavuconazole, and terbinafine from Brazil due to lack of inclusion of these antifungals in the survey. *, $P < 0.05$ - χ test based; full details and comparisons are presented in Table S1. Data from Brazil were previously included in a collective publication alongside data from other Caribbean and Latin American countries (26). The European data, in its present format, have been published separately elsewhere. (29)

Nevertheless, it is important to note that these disparities may be influenced by not only financial considerations, particularly in the case of liposomal amphotericin B, isavuconazole, and new formulations of posaconazole (such as delayed-release tablets and intravenous infusion), but also the absence of licensing for certain antifungals. Finally, the substantial difference in access to TDM reflects a critical disparity in optimizing antifungal treatment regimens between the two regions, hampering adherence to global standards (6, 41–47). Of note, the WHO's list of essential medicines includes some systemic antifungal drugs (48), such as amphotericin B deoxycholate and its liposomal formulation, anidulafungin, caspofungin, micafungin, fluconazole, itraconazole, voriconazole, and flucytosine. With the exception of amphotericin B deoxycholate, widely discouraged for use, and the non-mold-active fluconazole, the remaining antifungals listed are notably more prevalent in Europe compared with Brazil.

To overcome the drastic differences described above, further international collaboration programs should be set up, and those already active reinforced, encouraging collaborative efforts between Brazilian and European scientific organizations to facilitate knowledge exchange and capacity building. Furthermore, training programs must be established to enhance diagnostic and treatment capabilities in regions with identified gaps, including emerging technologies like PCR and MALDI-TOF-MS. Moreover, surveillance initiatives may need to track changes in IFI epidemiology and management practices over time, providing a more dynamic understanding of the situation. Also, advocacy for increased resource allocation in regions with identified disparities, addressing infrastructure needs and ensuring equitable access to diagnostic tools and antifungal medications. Finally, enhancement of public health awareness campaigns is pivotal with specific fungal pathogen prevalence, aiding in early detection and prevention. Some efforts had been made in this direction: The Brazilian government had recently expanded the antifungal armamentarium available in the public health system, diagnostic tests are increasingly available (especially for HIV-associated diseases), and there is increasing involvement of scientific societies and other stakeholders in these processes. (49)

While the study provides valuable insights, it is essential to acknowledge certain limitations. The significant time gap between the data collection periods in Brazil (2018) and Europe (2021–2022) may introduce biases in responses due to changes in context, such as healthcare policies and disease prevalence; however, given the slower pace of advancements in medical mycology, the impact of this limitation may be less pronounced. The data primarily rely on survey responses, introducing the possibility of reporting bias. It is also important to note that Brazil, like Europe, is a vast geographical area with great variability in economic factors, available resources, and even the incidence of mycoses within the country. Although the European survey was stratified by gross domestic product (GDP), the Brazilian survey was not, which could lead to bias. Despite these limitations, the overall comparative analysis offers a robust foundation for understanding regional differences in IFI management.

In conclusion, the study underscores the importance of tailored approaches to IFI management based on regional epidemiological and healthcare nuances. The identified gaps in diagnostic and treatment practices call for targeted interventions, knowledge exchange, and capacity building, particularly in regions with limited resources. As fungal infections continue to evolve, ongoing collaborative efforts between regions can facilitate a more holistic global approach to IFI prevention, diagnosis, and treatment.

## ACKNOWLEDGMENTS

The authors thank all participating institutions for their utmost contributions and support to the project during a pandemic situation and to all the individuals that have disseminated the link to the survey. Part of these results have been previously presented as an oral presentation by Dr. Diego R. Falci at 21st INFOCUS - Iguazu Falls - Brazil - November 2023.

D.R.F. and A.C.P. were in charge of the data collection for Brazil. O.A.C. and J.S.G. were in charge of the data collection for Europe. All authors conceived the study idea, collected data, and reviewed the manuscript. J.S.-G. wrote the initial version of the manuscript. All authors reviewed and accepted the final version of this manuscript.

## AUTHOR AFFILIATIONS

[1]Institute of Translational Research, Cologne Excellence Cluster on Cellular Stress Responses in Aging-Associated Diseases (CECAD), University of Cologne, University Hospital Cologne, Cologne, Germany

[2]Department I of Internal Medicine, Center for Integrated Oncology Aachen Bonn Cologne Duesseldorf (CIO ABCD) and Excellence Center for Medical Mycology (ECMM), University of Cologne, University Hospital Cologne, Cologne, Germany

³German Centre for Infection Research (DZIF), Partner Site Bonn-Cologne, Cologne, Germany

⁴Pontificia Universidade Católica do Rio Grande do Sul, Porto Alegre, Brazil

⁵Infectious Diseases Service, Hospital de Clinicas de Porto Alegre, Porto Alegre, Brazil

⁶Clinical Trials Centre Cologne (ZKS Köln), University of Cologne, University Hospital Cologne, Cologne, Germany

⁷Santa Casa de Misericordia de Porto Alegre, Porto Alegre, Brazil

⁸Federal University of Health Sciences of Porto Alegre, Porto Alegre, Brazil

## AUTHOR ORCIDs

Jon Salmanton-García http://orcid.org/0000-0002-6766-8297
Diego R. Falci http://orcid.org/0000-0002-8683-3833
Oliver A. Cornely http://orcid.org/0000-0001-9599-3137
Alessandro C. Pasqualotto http://orcid.org/0000-0002-6782-5395

## AUTHOR CONTRIBUTIONS

Jon Salmanton-García, Conceptualization, Data curation, Formal analysis, Investigation, Methodology, Project administration, Supervision, Validation, Visualization, Writing – original draft, Writing – review and editing | Diego R. Falci, Conceptualization, Investigation, Methodology, Supervision, Writing – review and editing | Oliver A. Cornely, Funding acquisition, Investigation, Methodology, Supervision, Writing – review and editing | Alessandro C. Pasqualotto, Conceptualization, Investigation, Methodology, Supervision, Writing – review and editing.

## DATA AVAILABILITY

Data will be available upon reasonable request to corresponding author.

## ADDITIONAL FILES

The following material is available online.

### Supplemental Material

**Table S1 (Spectrum02112-24-s0001.docx).** Access to treatment tools and therapeutic drug monitoring of participating institutions in Brazil and Europe.

### Open Peer Review

**PEER REVIEW HISTORY (review-history.pdf).** An accounting of the reviewer comments and feedback.

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
