## [Reviewer comments · Microbiology Spectrum]

Microbiology Spectrum

Elevating Fungal Care: Bridging Brazil's Healthcare Practices to Global Standards

Jon Salmanton-García, Diego Falci, Oliver Cornely, and Alessandro Pasqualotto

Corresponding Author(s): Jon Salmanton-García, Universitätsklinikum Köln Klinische Infektiologie

Review Timeline:

Submission Date:	August 22, 2024
Editorial Decision:	October 10, 2024
Revision Received:	October 14, 2024
Accepted:	October 16, 2024

Editor: Paschalis Vergidis

Reviewer(s): Disclosure of reviewer identity is with reference to reviewer comments included in decision letter(s). The following individuals involved in review of your submission have agreed to reveal their identity: Ariful Basher (Reviewer #4)

Transaction Report:

DOI: <https://doi.org/10.1128/spectrum.02112-24>

Re: Spectrum02112-24 (Elevating Fungal Care: Bridging Brazil's Healthcare Practices to Global Standards)

Dear Dr. Jon Salmanton-García:

Thank you for the privilege of reviewing your work. Below you will find instructions from the Spectrum editorial office and the reviewer comments.

Revision Guidelines

Sincerely,
Paschalis Vergidis
Editor
Microbiology Spectrum

Reviewer #1 (Comments for the Author):

An adequate information search strategy and the respective statistical analysis were applied.
What is the period of the information search? Is it the same period for all countries?

Reviewer #2 (Comments for the Author):

The authors have summarized the findings of a survey regarding mycological epidemiology and diagnostic and therapeutic possibilities in Brazil compared to Europe. As evidence regarding this topic specifically lacking for Brazil and the South-American part of the world, with on the other hand emerging fungal infections in that area, these results are of interest for the field. Although only descriptive results from a survey, conclusions regarding technical and therapeutic gaps (e.g. availability of molecular testing, *Candida*/*Mucor*/*Aspergillus* PCR) can have a clinical impact as they lead to future goals for the mycological field in Brazil. Some suggestions for future improvement regarding this topic are recommended in the Discussion.

Minor Remarks:

- Maybe interesting - As *Aspergillus* PCR is less available in Brazil and would be useful for detection of azole resistance - to mention in the Discussion some epidemiology or known information (are that information is lacking) on azole resistance rate in Brazil.
- The European results, are these the results published before as (Salmanton-García J, et al. *The Lancet Microbe* 2023; 4:e47-e56). If so, please mention more clearly in the methods/discussion.
- Line 323: "Underscored a significant gap", is more wording as for in the Discussion not the Result section.

Reviewer #3 (Comments for the Author):

Thanks for discussing such an interesting area. Please compare some data from Asia in the discussion section to provide a comprehensive assessment of why this technology is not popular despite a good GDP. The author can touch on why the gap is created is it a lack of knowledge or practice or legal practice? The epidemiology of fungal infection data of these two regions can be compared if any may be described that will help to alleviate the study limitations.

1 Dear Editor,

2

3 Following your editorial decision, please find enclosed the revised version of our paper
4 Spectrum02112-24 entitled 'Elevating Fungal Care: Bridging Brazil's Healthcare Practices to
5 Global Standards' that we revised according to the reviewers' helpful suggestions. We appreciated
6 the positive comments from the reviewers. We are grateful to the referees and to the Editorial staff
7 for your efforts in providing this review, which significantly improved the quality of our paper.

8

9 We read with great attention all the stimulating suggestions and accordingly we modified our
10 manuscript, following each point raised and highlighting with track changes the corrections in the
11 text.

12

13 **Reviewer #1**

14

15 1. An adequate information search strategy and the respective statistical analysis were applied.

16

17 Answer: We thank the reviewer for the positive feedback.

18

19 2. What is the period of the information search? Is it the same period for all countries?

20

21 Answer: We thank the reviewer for their insightful comment. As outlined in the Methods section,
22 data were collected in Brazil in 2018 and in Europe in 2021-2022. Additionally, we have
23 acknowledged this limitation as follows: "The significant time gap between the data collection
24 periods in Brazil (2018) and Europe (2021-2022) may introduce inconsistencies in responses due
25 to changes in context, such as healthcare policies and disease prevalence; however, given the
26 slower pace of advancements in medical mycology, the impact of this limitation may be less
27 pronounced."

28

29 **Reviewer #2**

30

31 1. The authors have summarized the findings of a survey regarding mycological epidemiology
32 and diagnostic and therapeutic possibilities in Brazil compared to Europe. As evidence
33 regarding this topic specifically lacking for Brazil and the South-American part of the world,
34 with on the other hand emerging fungal infections in that area, these results are of interest for
35 the field. Although only descriptive results from a survey, conclusions regarding technical and
36 therapeutic gaps (e.g. availability of molecular testing, *Candida/Mucor/Aspergillus* PCR) can

37 have a clinical impact as they lead to future goals for the mycological field in Brazil. Some
38 suggestions for future improvement regarding this topic are recommended in the Discussion.

39
40 Minor Remarks:

41
42 2. Maybe interesting - As *Aspergillus* PCR is less available in Brazil and would be useful for
43 detection of azole resistance - to mention in the Discussion some epidemiology or known
44 information (are that information is lacking) on azole resistance rate in Brazil.

45
46 Answer: We very much appreciate this comment and included a mention to the test in the
47 discussion.

48
49 3. The European results, are these the results published before as (Salmanton-García J, et al.
50 The Lancet Microbe 2023; 4:e47-e56). If so, please mention more clearly in the
51 methods/discussion.

52
53 Answer: We have indicated this in Methods, as follows: "Individual data for Brazil and Europe
54 were published separately in 2019 and 2023, respectively".

55
56 4. Line 323: "Underscored a significant gap", is more wording as for in the Discussion not the
57 Result section.

58
59 Answer: We agree and have removed such statement.

60
61 **Reviewer #3**

62
63 1. Thanks for discussing such an interesting area. Please compare some data from Asia in the
64 discussion section to provide a comprehensive assessment of why this technology is not
65 popular despite a good GDP. The author can touch on why the gap is created is it a lack of
66 knowledge or practice or legal practice? The epidemiology of fungal infection data of these
67 two regions can be compared if any may be described that will help to alleviate the study
68 limitations.

69
70 Answer: We celebrate this comment from the reviewer. We have now explicitly mention that the
71 situation in Brazil might be comparable to other world regions, like Africa or Asia/Pacific. Besides,
72 we propose "To overcome the drastic differences described above, further international
73 collaboration programs should be set up, and those already active reinforced, encouraging

74 collaborative efforts between Brazilian and European scientific organizations to facilitate
75 knowledge exchange and capacity building. Furthermore, training programs must be established
76 to enhance diagnostic and treatment capabilities in regions with identified gaps, including
77 emerging technologies like PCR and MALDI-TOF-MS. Moreover, surveillance initiatives may
78 need to track changes in IFI epidemiology and management practices over time, providing a
79 more dynamic understanding of the situation. Also, advocacy for increased resource allocation in
80 regions with identified disparities, addressing infrastructure needs and ensuring equitable access
81 to diagnostic tools and antifungal medications. Finally, enhancement of public health awareness
82 campaigns is pivotal with specific fungal pathogen prevalence, aiding in early detection and
83 prevention.”

84

85 We thank you for having given us the opportunity of revising and improving the manuscript.
86 Hoping that the revised version of our work might be suitable for publication in Microbiology
87 Spectrum, we send our best regards.

88

89 Sincerely yours,

90 Jon Salmanton-García

Re: Spectrum02112-24R1 (Elevating Fungal Care: Bridging Brazil's Healthcare Practices to Global Standards)

Dear Dr. Jon Salmanton-García:

You may clarify the following points for the publication:

-Lines 304-305: "The Brazilian survey involved 96 institutions and European survey had 388 participants". How many European institutions were involved?

-Lines 345-347: "Accessibility to at least one triazole did not exhibit significant difference. However, all triazoles except fluconazole were more commonly at hand in Europe". You may consolidate this statement (No difference for fluconazole. Other azoles were more accessible in Europe).

Your manuscript has been accepted, and I am forwarding it to the ASM production staff for publication. Your paper will first be checked to make sure all elements meet the technical requirements. ASM staff will contact you if anything needs to be revised before copyediting and production can begin. Otherwise, you will be notified when your proofs are ready to be viewed.

Sincerely,
Paschalis Vergidis
Editor
Microbiology Spectrum